# Effects of *Lactobacillus* on the Differentiation of Intestinal Mucosa Immune Cells and the Composition of Gut Microbiota in Soybean-Sensitized Mice

**DOI:** 10.3390/foods12030627

**Published:** 2023-02-01

**Authors:** Chunhua Yang, Jierui Zhu, Jing Bai, Jie Zhang, Zhihua Wu, Xin Li, Ping Tong, Hongbing Chen, Anshu Yang

**Affiliations:** 1State Key Laboratory of Food Science and Technology, Nanchang University, Nanjing Dong Lu 235, Nanchang 330047, China; 2Sino-German Joint Research Institute, Nanchang University, Nanjing Dong Lu 235, Nanchang 330047, China; 3Experimental Animal Center, Jiangxi University of Chinese Medicine, 1688 Meiling Road, Nanchang 330004, China

**Keywords:** *Lactobacillus*, probiotics, intestinal mucosal immunity, immune-cell differentiation, gut microbiota, soybean allergy

## Abstract

In the early stage of this study, three strains of *Lactobacillus* with anti-soybean allergy potential were screened: *Lactobacillus acidophilus* CICC 6081, *Lactobacillus delbrueckii* subsp. *Bulgaricus* CICC 6103 and *Lactobacillus plantarum* subsp. *Plantarum* CICC 20988. The aim of this study was to analyze the desensitization effect of three strains of *Lactobacillus* administered by gavage to soybean-allergic mice through the differentiation of immune cells in intestinal lymph nodes and the changes to gut microbiota. The results showed that the three strains of *Lactobacillus* could stimulate the proliferation of dendritic cells (DCs) and regulate the balance of Th1/Th2 differentiation in the MLNs and PPs of soybean-allergic mice. Furthermore, the Th17/Tregs cell-differentiation ratio in the MLNs of the *Lactobacillus*-treated mice was significantly lower than that of the allergic mice (*p* < 0.05). Compared to the control group, the Shannon, Sobs and Ace indexes of intestinal microbiota in the allergic mice were significantly increased (*p* < 0.05), and the proportion of *Clostridiales* was significantly higher (*p* < 0.05), which was reversed by *Lactobacillus* gavage. In conclusion, the three strains of *Lactobacillus* can inhibit the intestinal mucosal immune response and regulate gut microbiota balance in soybean-allergic mice.

## 1. Introduction

Soybean is one of the main allergenic foods, and susceptible people will experience IgE-mediated type I hypersensitivity when they are accidentally exposed to soybean. The patient has allergic dermatitis, respiratory symptoms, gastrointestinal dysfunction and other symptoms [1]. There are 40 studies that have determined the weighted prevalence of soy allergies; for the general population, about 0.3% of people suffer from a soybean allergy, while a soy allergy is more common in allergic children, with an average prevalence of 2.7%, and some studies have reported it to be as high as 12.9% [2]. These allergy diagnosis cases were based on clinical manifestations, patient reports, serum concentrations of sIgE, SPT or an oral challenge test [2]. Clinically, several allergen immunotherapies (oral immunotherapy, sublingual immunotherapy and epicutaneous immunotherapy) are used to establish allergen tolerance [3], but strict avoidance of soy is the best way to prevent allergic reaction. However, soybean has become a common food and food ingredient all over the world. Soybean is not only a traditional food in the Eastern diet, but nowadays, soy-based food additives are also commonly found in the Western diet [4], which has increased the incidence rate of soybean allergy in adults. For the patients with a soybean allergy, this will undoubtedly seriously affect their normal life. Methods of treating and preventing soybean allergic reaction in patients has become a direction of exploration for allergy researchers.

The contact between the human body and most food allergens occurs on the intestinal mucosa. Mesenteric lymph nodes (MLNs) and Peyer’s patches nodes (PPs) are thought to induce adaptive immunity as induction sites of the intestinal immune system [5]. PPs are the early and direct sites for inducing the intestinal epithelial immune response, and it is also the sites of antigen uptake [6]. In the presence of allergic reaction, dendritic cells (DCs) can ingest and process exogenous antigens (including soybean protein), which bind to MHC II molecules and present them to naive T cells through MHC II, thus initiating an immune response through the MHC II pathway [7]. Due to the central role of classical or conventional DCs (cDCs) in T cell activation, they have been implicated in the development of T-cell-mediated autoimmune, inflammatory and allergic diseases [8]. Among them, CD103^+^ DCs could highly express the RALDH2 enzyme, which can metabolize retinal to retinoic acid (RA) [9]. The high expression of TGF-β by CD103^+^ DCs can induce Tregs cells, and this induction can be enhanced by RA [9]. In addition, RA can induce intestinal homing receptors and integrin α4β7 [10,11], which induce the differentiation of naive T cells into Tregs. The induction of oral tolerance and its systemic effect are mainly caused by the action of Tregs produced by antigens stimulation in the intestinal mucosa [12]. Although the mechanisms by which Tregs affect the whole body are not known, there are many reports about the sites in the intestine that are involved in the induction of oral tolerance, for example MLNs [13] and PPs [14]. Therefore, it is necessary to investigate the differentiation of CD103^+^ DCs and CD4^+^ T cell subpopulations in MLNs and PPs.

Some scholars have thought that the gut microbiota is an important “invisible organ” of the human body [15], constituting a complex ecosystem that promotes host function. For example, micro-organisms decompose carbohydrates into short-chain fatty acids and use tryptophan to produce indoxyl metabolites, which can help shape the function of innate and adaptive immune cells [16]. As the largest immune system in vertebrates, the intestinal mucosal immune system is closely related to the gut microbiota. The rapid colonization of microbiota in the gastrointestinal tract of neonates plays a crucial role in the development of the intestinal immune system; in contrast, studies on germ-free animals have shown that the loss of gut microbiota can lead to severe immune system deficiencies [17]. In addition, the imbalance of gut flora can lead to autoimmune, allergic and chronic inflammatory diseases [18,19,20]. Researchers have reported that germ-free mice were more susceptible to allergic reactions induced by oral antigens than normal animals, so the intestinal microbiome as a whole can provide signals to inhibit food allergy, thus protecting the host from allergy [21].

Probiotics are kinds of non-pathogenic micro-organisms beneficial to human health [22]. In daily life, people usually get them through diet (fermented products include yogurt, douchi, etc., or isolated probiotic strains). Appropriate probiotics can help to establish the balance of gut microbiota and promote the development of lymphoid tissues in the intestinal mucosa [23]. In terms of regulating the host immune system, the interaction between probiotics and local (intestine, vagina, urethra) host immune cells can induce the production of cytokines [24]. *Lactobacillus* are commonly used probiotics that can theoretically prevent or suppress immune diseases [25]. The exact mechanism of *Lactobacillus* in alleviating food allergies is still a mystery, but some studies have suggested potential mechanisms, such as the regulation of Th1/Th2 balance, the enhancement of the intestinal barrier and the production of short-chain fatty acids. [21]. A large number of studies have proved that *Lactobacillus* have a positive effect on the prevention and treatment of food allergies. Some *Lactobacillus* can alleviate the allergic reactions caused by bovine lactoglobulin, ovalbumin and so on [26,27,28]. A food allergy can disrupt the balance of Th1/Th2, making them biased toward Th2. Elen [29] found that *Lactobacillus bulgaricus* can regulate the differentiation of Th1/Th2 cells to alleviate the allergic asthma inflammation caused by OVA. Other studies have shown that some *Lactobacillus* can reduce allergic reactions by regulating Th1/Th2 balance, enhancing the intestinal barrier function and regulating the gut microbiota composition [30]. However, for the research on soybean and *Lactobacillus*, more attention has been paid to the effect on the quality and function of soymilk or soybean after fermentation with *Lactobacillus*, while the studies on *Lactobacillus* interfering with soy protein allergies are limited. In addition, many clinical experiments have demonstrated that the effect of *Lactobacillus* on food allergy is strain-specific [31]. Therefore, further studies are needed to determine which exogenous *Lactobacillus* can alleviate soybean allergy by affecting the gut microbiota of soy-allergic mice.

Taken together, in this study, three strains of *Lactobacillus* with good anti-soy allergy properties previously screened by our research team were selected [32], namely *Lactobacillus acidophilus* CICC 6081 (La), *Lactobacillus delbrueckii* subsp. *Bulgaricus* CICC 6103 (Ld) and *Lactobacillus plantarum* subsp. *Plantarum* CICC 20988 (Lp). They belong to *Lactobacillus* genus and are usually used in functional foods and food supplements. Then, a soybean-protein-sensitized mouse model was established, and MLNs, PPs and cecum were isolated from the intestine of mice. The aim of this study was to evaluate the effect of *Lactobacillus* intervention on soy-allergic mice from the perspective of immune cells and gut microbiota in intestinal lymph nodes.

## 2. Materials and Methods

### 2.1. Materials

Soybean cultivar (Dong Nong 48) was provided by Northeast Agricultural University, China. *Lactobacillus acidophilus* CICC 6081, *Lactobacillus delbrueckii* subsp. *bulgaricus* CICC 6103 and *Lactobacillus plantarum* subsp. *Plantarum* CICC 20988 were obtained from China Center of Industrial Culture Collection. Cholera toxin was received from Sigma-Aldrich (St. Louis, MO, USA). Cell Stimulation Cocktail plus protein transport inhibitors (500×) was acquired from eBioscience (Waltham, MA, USA). Anti-Rat and Anti-Hamster Igκ/Negative Control Compensation Particles Set, Transcription Factor Buffer Set, BD Cytofix/Cytoperm™ Fixation/Permeabilization Kit, APC-anti-mouse CD3e antibody, PE-anti-mouse CD103 antibody, FITC-anti-mouse CD4 antibody, APC-anti-mouse CD11c antibody, PE-anti-mouse CD8a antibody, PE-anti-mouse IL-4 antibody, PE-anti-mouse CD25 antibody, Perep-Cy5.5-anti-mouse IFN-γ antibody, Aleax 647-anti-mouse Foxp3 antibody and PE-anti-mouse IL-17A antibody were all purchased from Becton, Dickinson and Company (Franklin Lakes, NJ, USA). All other chemicals were of analytical grade.

### 2.2. Preparation of Lactobacillus Solution

The three strains of *Lactobacillus* stored at −80 °C were activated at suitable temperature for 24 h. The activated bacteria were inoculated in fresh de Man, Rogosa and Sharpe Broth and expanded to a stable stage at 37 °C. The bacterial suspension was plated on a Petri dish containing solid medium for plate counting. Then, the bacteria were collected by centrifugation (3000× *g*, 5 min, 4 °C) and washed twice with sterile PBS. Then, the bacteria were suspended in sterile PBS and the bacterial concentration was adjusted to 5 × 10^9^ CFU/mL. The *Lactobacillus* suspension required for the intragastric administration of mice was obtained, and stored at 4 °C.

### 2.3. Preparation of Protein Solution

Soybean was soaked overnight (soybean/distilled water = 1:6.5, *m*/*v*); then, soybean and 6-fold of water were ground in a soymilk machine and filtered by gauze to obtain soymilk [33]. Soymilk and 0.05 M Tris-HCl solution were mixed in equal volume (pH 8.0), and then the double volume of n-hexane was added and stirred for 6 h. Soybean oil was removed from the upper layer by centrifugation (30 min, 8000× *g*) at 4 °C, and the protein solution was obtained from the lower layer solution by centrifugation again, and stored at −20 °C.

### 2.4. Establishment of Soybean ALLERGY Model in Mice

The sensitization protocol of the mice corresponded to the previous method and was slightly modified [34]. BALB/c mice were purchased from Beijing Vital River Laboratory Animal Technology Co., Ltd. All the mice used in this study were handled in accordance with the Animal Management regulations published by the Ministry of Health of the People’s Republic of China. All experimental procedures were approved by the Animal Care Review Committee of Jiangxi University of Chinese Medicine (ethical approval code: JZLLSC2019-0307). Twenty-one SPF BALB/c mice were bred with AIN93G standard feed without soybean meal in the SPF animal room of the Experimental Animal Science and Technology Center of Jiangxi University of Traditional Chinese Medicine to breed the second-generation mice.

Thirty female second-generation mice aged 4–5 weeks (weight 15–20 g) were selected as experimental subjects. The sensitization scheme of the mice is shown in Figure 1. Mice were randomly divided into 5 groups (n = 6): blank control group (C), soybean protein group (SP), *Lactobacillus acidophilus* CICC 6081 group (La), *Lactobacillus plantarum* subsp. *Plantarum* CICC 20988 group (Lp) and *Lactobacillus delbrueckii* subsp. *Bulgaricus* CICC 6103 group (Ld). The groups of La, Ld and Lp were gavaged with 10^9^ CFU [35] of the bacteria strain every day, and the other two groups received equal doses of PBS by gavage. At week 0, 1, 2 and 3, each mouse in the experimental groups was gavaged with PBS containing 5 mg soy protein and 10 μg cholera toxin (CT). In the fourth week, each mouse was stimulated with 20 mg of soy protein by gavage, and C group was replaced with sterile PBS. After 40 min of challenge, the mice were anesthetized and dislocated to death. Then, the intestine was dissected under aseptic conditions, and MLNs, PPs and cecum were carefully separated. The contents of the cecum were placed in a cryotube and then immediately frozen in liquid nitrogen for later use.

### 2.5. Body Weight and Allergic Symptom Scores of Mice

The body weight of the mice was recorded before gavage on days 0, 7, 14, 21 and 28. After 40 min of soy protein challenge, the symptoms of each group of mice were scored according to the allergic symptom score table (Table 1).

### 2.6. Analysis of Immune Cell Differentiation in Intestinal Mucosal Immunity

#### 2.6.1. Preparation of Mouse MLN and PP Cell Suspension

The MLNs or PPs together with RPMI-1640 medium were poured into a 70 μm cell sieve, ground and sieved completely. The cell suspension was centrifuged at 1200 rpm for 5 min, and the supernatant was discarded. Then, RPMI-1640 medium was added to wash the cells and centrifuged again to discard the supernatant, and 1 mL of complete medium (RPMI-1640 medium containing 15% fetal bovine serum, 1% Penicillin-Streptomycin Liquid, and 1% glutamine) was added to resuspend the cells. The cells were mixed with Tissue Blue and counted on a cell-counting plate. The MLN and PP cell suspension of 500 μL (4 × 10^6^ cells/mL) of mice in each group was pipetted into a 48-well cell culture plate, and then cell stimulation cocktail plus protein transport inhibitors 500× (final concentration: 2 μL/mL) were added to each well and incubated at 37 °C and 5% CO_2_ for 16 h. The remaining cells were not stimulated, and the concentration was adjusted to 10^7^ cells/mL with stain buffer.

#### 2.6.2. Identification of DCs and CD4^+^/CD8^+^ T Cells in MLNs and PPs

Unstimulated MLN and PP cells (10^6^ cells) were added to the 96-well tip plate, and blank control well (C group mouse cell suspension) was set at the same time. For the DCs assay, APC-anti-mouse CD11c antibody and PE-anti-mouse CD103 antibody were added into sample wells, while the antibodies added for the CD4^+^/CD8^+^ T cells assay were APC-anti-mouse CD3e antibody, FITC-anti-mouse CD4 antibody and PE-anti- mouse CD8a antibody. Then, the cells were incubated at 4 °C in dark for 30 min. After washing twice with stain buffer, the cells were resuspended, filtered with nylon mesh and added into the flow tube. Then, each sample was detected with a BD FACSVerse flow cytometer.

#### 2.6.3. Detection of IFN-γ and IL-4 in MLNs and PPs

The percentage of Th1 and Th2 cells was reflected by the content of IFN-γ and IL-4, respectively. The stimulated cells (10^6^ cells) were laid on the 96-well tip plate. The FITC-anti-mouse CD4 antibody was added into each sample wells with the exception of the blank control well, and incubated at 4 °C for 30 min in dark. After washing twice with stain buffer, the fixation/permeabilization solution was added into each well and incubated at 4 °C for 20 min in the dark. The cells were washed twice with BD Perm/wash buffer, and percp-cy5.5-anti-mouse IFN-γ antibody and PE-anti-mouse IL-4 antibody were added into each sample well and incubated at 4 °C for 30 min. Then, the cells were washed with BD Perm/wash buffer, and IFN-γ and IL-4 were detected according to the method in Section 2.6.2.

#### 2.6.4. Identification of Th17 Cells and Tregs in MLNs and PPs

The stimulated cells (10^6^ cells) were laid on the 96-well tip plate. The FITC-anti-mouse CD4 antibody was added into each sample well with the exception of the black control well, and PE-anti-mouse CD25 antibody was added to the sample wells for Tregs measurement. Then, the cells were incubated at 4 °C for 30 min in dark. After washing twice with stain buffer, fixation/permeabilization solution was added into each well and incubated at 4 °C for 20 min in dark. The cells were washed twice with perm/wash buffer. Then, perm/wash buffer was added into each well for resuspension, PE-anti-mouse IL-17A antibody was added into Th17 sample wells and Aleax 647-anti-mouse Foxp3 antibodies were added into Tregs sample wells. The cells were incubated at 4 °C for 30 min. Then, the cells were washed and resuspended with perm/wash buffer, filtered with nylon mesh and added into the flow tube. Lastly, each sample was detected with a BD FACSVerse flow cytometer.

### 2.7. Microbial Composition Analysis of Cecal Contents in Mice

Genomic DNA was extracted from the cecal contents. The 16S rRNA gene amplicon was sequenced on the Illumina MiSeq platform, and the primer sequences was derived from the V3-V4 high variant region of the bacterial 16S rRNA gene (338F = 5′-ACTCCTACGGGAGGCAGCAG-3′, 806R = 5′-GGACTACHVGGGTWTCTAAT-3′). All sequences reported here have been deposited in the NCBI Sequence short read archive (Accession to cite for these SRA data: PRJNA906639). Then, the sequencing data were spliced, quality-controlled and de-spliced to obtain the optimized sequences. OTU clustering was performed on the optimized sequences to obtain OTU abundance tables. After the OTUs were drawn flat based on the minimum number of sequences, analyses for α diversity and β diversity were performed. The data were analyzed on the free online platform Majorbio Cloud Platform (https://www.majorbio.com/ accessed on 25 November 2022).

### 2.8. Statistical Analysis

Flow cytometry data were plotted by FlowJo 10.6.1 software. The related data of microbial diversity were plotted and analyzed by Meggie cloud biological platform. Results were plotted and analyzed by software GraphPad Prism 8. First, it was used to perform normality tests and checked by the Shapiro–Wilk test. Then, the one-way ANOVA was used based on testing, and the Tukey test was used for multiple comparisons between the different treatment groups. The results were expressed as mean ± S.D., and different letters denote statistically significant differences (*p* < 0.05).

## 3. Results

### 3.1. Allergic Response in Mice

It was found from Figure 2 that the body-weight gain trend of the SP group was slower than that of the C group. The trend of body weight gain in the Lp and Ld groups was similar to that in the C group, while the body weight of the La group increased faster than that of the C group from day 7 to day 14. Moreover, as shown in Table 2, all mice in the experimental groups, except for the blank control group, showed allergic symptoms with different degrees after challenge. The mice of the SP group had allergic symptoms such as scratching the nose or head and accelerated breathing and diarrhea, while only some mice in the *Lactobacillus* groups had mild symptoms and scratching of the nose.

### 3.2. Percentage of DCs in MLNs and PPs

The changes of CD103^+^ DCs percentage in the MLNs and PPs of the mice gavaged with *Lactobacillus* is shown in Figure 3. There were no significant differences between the groups. The percentage of CD103^+^ DCs in the MLNs and PPs of the SP group was decreased compared with C group. It showed that sensitization with soybean protein could inhibit the differentiation of CD103^+^ DCs in the MLNs and PPs of mice. The percentage of CD103^+^ DCs in *Lactobacillus* groups were all higher than that in the SP group. In addition, the three strains of *Lactobacillus* had the same effect, which could make the differentiation level of CD103^+^ DCs in the MLNs and PPs slightly higher than that in the SP group. Although there were no significant differences, it is not difficult to see that the effect of La (*p* < 0.1) and Ld (*p* < 0.1) is better than that of Lp (*p* > 0.1) in PPs.

### 3.3. Percentage of CD4^+^ and CD8^+^ Cells in MLNs and PPs

The activation and differentiation of DCs will directly affect the differentiation of T lymphocytes in vivo. Therefore, flow cytometry was used to further identify the difference in the expression levels of CD4^+^ (Th cells) and CD8^+^ (Tc cells) in MLNs and PPs. As shown in Figure 4, the percentage of CD4^+^ T cells in MLNs and PPs was as high as 70% or more, while the percentage of CD8^+^ T cells was less than 30%, indicating that T cells in the MLNs and PPs of mice are mainly Th cells. Compared with C group, CD4^+^ T cells in the MLNs (*p* < 0.05) and PPs (*p* = 0.11) of the SP group mice increased, while CD8^+^ T cells showed a downward trend, indicating that soybean protein sensitization could promote the differentiation of T cells into Th cells in the MLNs and PPs of mice. Compared with the SP group, the percentage of CD4^+^ T cells in the MLNs and PPs of the three *Lactobacillus* groups all decreased to varying degrees, while the percentage of CD8^+^ T cells increased, but there were no significant differences. These results showed that the three strains of *Lactobacillus* could inhibit the differentiation of T cells to Th cells in the MLNs and PPs of soybean-protein-allergic mice and promote their differentiation to Tc cells.

### 3.4. Differentiation Balance of Th1 and Th2 Cells in MLNs and PPs

Th cells activated by antigens presented by MHCII can secrete cytokines that regulate or assist in immune responses. Nowadays, there is no surface marker to distinguish Th1 and Th2 cells, so the content of intracellular characteristic cytokine IFN-γ was used to reflect the percentage of Th1 cells, and the content of IL-4 was used to reflect the percentage of Th2 cells [36]. The changes in the percentage of Th1 cells and Th2 cells in the MLNs of the mice were shown in Figure 5A,B. The percentage of Th1 cells in the MLNs of the SP group decreased to 14.8%, which was significantly different from that of the C group (*p* < 0.05). However, the three strains of *Lactobacillus* promoted the differentiation of Th1 cells in the MLNs of soybean-protein-allergic mice to varying degrees (La 17.8%, Lp 18.0% and Ld 19.0%) compared to the SP group, and the effect in the Ld group was the most significant (*p* < 0.05) (Figure 5A). Conversely, soybean protein caused a significant increase in the percentage of Th2 cells in the MLNs of the mice (*p* < 0.01), and all three strains of *Lactobacillus* significantly reduced the percentage of Th2 cells in the MLNs of the allergic mice, among which Ld had the most significant effect (*p* < 0.01) (Figure 5B). In summary, soybean protein sensitization can inhibit the differentiation of Th1 cells and stimulate the differentiation of Th2 cells in the MLNs of mice, while *Lactobacillus* gavage can reverse this phenomenon.

The changes in the percentage of Th1 cells and Th2 cells in the PPs of the mice are shown in Figure 5C,D. Similar to the differentiation of Th1 and Th2 cells in MLNs, the percentage of Th1 cells decreased, while the percentage of Th2 cells increased in the PPs of soybean-sensitized mice. *Lactobacillus* gavage can stimulate the differentiation of Th1 cells to varying degrees, and inhibit the differentiation of Th2 cells.

The ratio of Th1/Th2 in the MLNs and PPs of the mice in each group was further analyzed. As shown in Figure 5E,F, compared with non-sensitized mice, the Th1/Th2 balance of MLNs and PPs in the sensitized mice were shifted toward Th2, with a significant shift in MLNs. Compared with the SP group, the ratio of Th1/Th2 in the MLNs was significantly increased in the *Lactobacillus* groups (*p* < 0.05), while the ratio of Th1/Th2 in the PPs was also slightly increased.

### 3.5. Differentiation Balance of Th17 Cells and Tregs in MLNs and PPs

Intracellular IL-17A was used as a marker to identify Th17 cells in the intestines of the mice. The percentage of CD4^+^ IL-17A^+^ cells in the MLNs was shown in Figure 6A. However, the CD4^+^ IL-17A^+^ cells were not identified in the PPs, which may be due to the absence of Th17 cells in the PPs or the fact that its content was too low to be detected. It can be seen from Figure 5 that the percentage of CD4^+^ IL-17A^+^ cells in the MLNs of the soybean-protein-sensitized mice significantly increased from 0.99% to 1.95% (*p* < 0.05). *Lactobacillus* gavage significantly reduced the percentage of CD4^+^ IL-17A^+^ cells in the MLNs of soybean-protein-sensitized mice (La 1.18%, Lp 1.24% and Ld 1.15%, *p* < 0.05), indicating that these three strains of *Lactobacillus* could inhibit the differentiation of Th17 cells in the MLNs of soybean-protein-allergic mice. Obviously, Ld has the most significant inhibitory effect on the differentiation of Th17 cells in the MLNs.

Tregs are also closely related to Th1/Th2 balance. In allergic subjects, the percentage of Tregs will decrease, leading to the development of the Th1/Th2 balance towards Th2 immunity [37]. Conversely, when the percentage of Tregs increases, it can regulate the balance of Th1/Th2 cells and maintain immune tolerance. Figure 6B,C shows the changes of the Tregs percentage in the MLNs and PPs of the mice. The percentage of Tregs in the MLNs of the SP group was significantly lower than that of the C group (*p* < 0.01), while the percentages in the La, Lp and Ld groups were higher than that of the SP group. Among them, the La and Ld groups were significantly different from the SP group (*p* < 0.05). The percentage of Tregs in the PPs was similar to that in the MLNs. In other words, soybean-protein sensitization could inhibit the differentiation of Tregs in the PPs, and the three strains of *Lactobacillus* could stimulate the differentiation of Tregs in the PPs of the soybean-protein-sensitized mice to a certain extent. In conclusion, the three strains of *Lactobacillus* can promote the differentiation of Tregs in the MLNs and PPs of the soybean-protein-allergic mice to varying degrees, which is consistent with the above results of stimulating the differentiation of Th1 cells and inhibiting the differentiation of Th2 cells. The percentage of Tregs in the MLNs and PPs of the Ld group was closer to that of the C group than that of the other two groups.

In addition to Th1/Th2, the ratio of Th17/Tregs is also noteworthy. In Figure 6D, the Th17/Tregs in the MLNs of the mice in each group was demonstrated, and Th17/Tregs in the PPs was not analyzed because Th17 was not detected in the PPs. The Th17/Tregs balance in the MLNs of soybean-allergic mice was significantly shifted to Th17 (*p* < 0.05). However, this bias was reversed again in the *Lactobacillus* groups compared to the SP group, which was similar to the change in the ratio of Th1/Th2.

### 3.6. Changes of Gut Microbiota in Mice

In this study, α- and β-diversity analyses of the gut microbiota composition of the five experimental groups were performed in depth. The α-diversity analysis included the Sobs index, Shannon index and Ace index, as shown in Figure 7A–C. The Sobs index can reflect the abundance of the community, the Shannon evenness measurement index can reflect the diversity of the community and the Ace index can estimate species diversity based on rare species. The three indices of the SP group were significantly higher than those of the C group (*p* < 0.05). However, compared with the SP group, the three indices of the *Lactobacillus* groups decreased to varying degrees, indicating that *Lactobacillus* could inhibit the elevation of microbial abundance and diversity in the intestine of mice caused by soy protein sensitization, and the effect of Lp was greater than that of the other two strains of *Lactobacillus*. Based on Bray–Curtis distance, PCoA analysis showed that the other groups were clearly distinguished compared to the SP group. The *Lactobacillus* groups significantly affected the distribution of microbial community in the caecum of soy-allergic mice (Figure 7D).

In order to further explore the differences in gut microbial community composition among different groups of mice, the species of intestinal microbiota in the different groups of mice were statistically analyzed. The composition of gut microbiota at the Order level is shown in Figure 7E, and the results of the cluster analysis at the Order level are shown in Figure 7F. The five most abundant Orders in the intestinal tract of the mice in each experimental group were *Clostridiales*, *Bacteroidales*, *Desulfovibrionales*, *Lactobacillales* and *Bacillales*, of which *Clostridiales* and *Bacteroidales* were dominant. Soybean protein sensitization caused a significant decrease in the proportion of *Clostridiales* in the intestinal tract of the mice (*p* < 0.05), while *Lactobacillus* gavage led to an increase in the proportion of *Clostridiales* in the intestinal tract of the sensitized mice. In contrast to *Clostridiales*, soybean-protein sensitization can cause a significant increase in the proportion of *Bacilales* (*p* < 0.05), while La, Lp and Ld can significantly reduce the proportion of *Bacilales* (*p* < 0.05). In other words, *Lactobacillus* had the ability to change the distribution of gut microbiota in mice and put them in a pre-allergenic state.

## 4. Discussion

The establishment of an animal model of soy allergy is the basis for exploring whether *Lactobacillus* can interfere with a food allergy. The results of the allergic response in the mice showed that the soybean-sensitized mouse model had been successfully established, and *Lactobacillus* could alleviate the slow weight gain and allergic symptoms caused by the allergy.

DCs are widely distributed in the intestine and gut-associated lymphoid tissues such as PPs and MLNs. In this study, soybean allergy could reduce the percentage of CD103^+^ DCs in the mice, as in other studies. Tan [38] showed that high fiber feeding mice can enhance the activity of retinal dehydrogenase in CD103^+^ DCs, thus enhancing oral tolerance and preventing food allergy. Similarly, Ma [39] reported that the probiotic mixtures inhibited OVA-induced anaphylaxis in a mouse model of food allergy, and the DCs in the PPs and MLNs of allergic mice increased dramatically after probiotics treatment. At the same time, our study found that *Lactobacillus* can promote the differentiation of CD103^+^ DCs, belonging to cDC1 in the intestinal tract of allergic mice to a certain extent. In mice, cDC1-mediated priming can promote the generation of Tregs and oral tolerance. DCs can regulate intestinal immune tolerance by promoting the differentiation of CD4^+^ T cells to Tregs and activating Tregs through a non-classical autophagic pathway [40]. The protective effect of probiotics depends partly on the maintenance of CD103^+^ DCs tolerance function. Wu [41] et al. also reported that retinoic acid signaling in splenic DCs can maintain Tregs in the spleen. In our results, the change trend of Tregs in the three *Lactobacillus* groups were positively correlated with the differentiation of DCs. Therefore, it is speculated that *Lactobacillus* may re-establish the immune tolerance to soybean protein by stimulating the activation of CD103^+^ DCs and maintaining Tregs. Interestingly, it was found that CD4^+^ T cells decreased and Th1 cells increased in the intestinal mucosa of the mice treated with *Lactobacillus*. Therefore, although CD103^+^ DCs could not stimulate the proliferation of CD4^+^ T cells, it could promote the differentiation of CD4^+^ T cells into Tregs and Th1 cells.

A large number of studies have shown that the destruction of the Th1/Th2 balance in an immune response is an important mechanism of food allergy. The production of antigen-specific IgE, the activation of intestinal mast cells and the enhancement of food antigen transport across intestinal epithelial cell are all controlled by Th2 cells. Wambrre [42] identified a unique subtype of antigen-specific Th2 cells in allergic patients, which can drive IgE class conversion and enhance the expansion of allergic effector cells. As a subset of T cells with positive synergistic effect with Th2 cells, Th17 cells can adversely affect a food allergy and play an important role in Th2-mediated pathogenesis [43]. Some studies have suggested that an imbalance of Th17/Tregs cells is associated with allergic reactions. For example, Shi [44] reported that RSV infection disrupted asthma tolerance by increasing the Th17/Tregs ratio rather than the Th1/Th2 ratio. Therefore, in order to comprehensively clarify the molecular and cellular mechanism of *Lactobacillus* against soybean allergy, we not only studied the differentiation balance of Th1/Th2 but also focused on the differentiation ratio of Th17/Tregs. The results showed that the three strains of *Lactobacillus* increased the ratio of Th1/Th2 and decreased the ratio of Th17/Tregs in soy-sensitized mice. It suggested that the *Lactobacillus* could maintain the balance of Th1/Th2 in the intestinal tract of the soy-allergic mice after intragastric administration. However, the ability of the *Lactobacillus* to maintain the differentiation of Th1/Th2 cells in the MLNs and PPs was different and was better in MLNs than PPs. Simultaneously, the effect of Ld was the best among the three strains. The *Lactobacillus* could stimulate the differentiation of Tregs in the MLNs of the soybean-protein-allergic mice, and the effects of La and Ld were better than that of Lp. These results further indicated that the *Lactobacillus* could alleviate a soybean protein allergy by affecting the differentiation of intestinal immune cells. Similarly, Zhang [45] showed that oral administration of *Clostridium butyricumo* CGMCC 0313-1 could reverse the imbalance of Th1/Th2 and Th17/Tregs by increasing the number of CD4^+^ CD25^+^ Foxp3^+^ Tregs, thus improving the symptoms of an intestinal allergic reaction in BLG-allergic mice. Liu et al. [46] also showed that *Bifidobacterium lactis* can effectively alleviate the allergic symptoms in children and mice, and reduce the ratio of Th17/Tregs. DCs have a significant polarization effect on Th cell differentiation, such as the production of interleukin-12 and the activation of STAT4 that can promote the polarization of Th1 cells. However, the BALB/c mouse model used in this study is dynamic and complex, which cannot exclude the influence of other conditions such as gut microbiota.

The “Hygiene Hypothesis” proposed the relationship between gut microbiota and human health. Since then, a large number of studies have shown that the gut microbiota has a variety of functions, such as increasing the intestinal mucosal barrier, promoting the maturation of the immune system, stimulating the secretion of intestinal IgA, preventing the adhesion of pathogenic microorganisms and participating in the formation of immune tolerance [21]. Supplementation with *Bifidobacterium* and *Lactobacillus*, the dominant bacteria in early infancy, can lead to beneficial changes in the gut microbiota of premature infants [47]. Exogenous probiotics can regulate the imbalance of gut microbiota caused by allergy so as to achieve the effect of preventing or alleviating allergic diseases. For example, the oral administration of *Lactobacillus Plantarum* ZDY 2013 and *Lactobacillus rhamnosus GG* can relieve the allergic reactions caused by β-lactoglobulin. Meanwhile, *Lactobacillus plantarum* ZDY 2013 can regulate the changes of gut microbiota caused by cow’s milk allergy [48]. Similarly, the results of this study showed that La, Lp and Ld could regulate the imbalance of intestinal microflora induced by soybean protein to some extent. The Sobs index, Shannon index and Ace index of the intestinal micro-organisms in the mice were significantly increased due to sensitization of soybean protein, indicating that soy protein sensitization led to an increase in the abundance and diversity of micro-organisms in the intestine of the mice, which is consistent with the results of Canani [49]. They found that the intestinal microflora of infants with a milk allergy were more diverse than that of healthy controls. This may be due to the destruction of the intestinal barrier in allergic mice. In this study, oral gavage of *Lactobacillus* could reverse this phenomenon to a certain extent and reduce the richness and diversity of the intestinal micro-organisms in mice sensitized by soybean protein. Further results of gut microbiota species showed that the increase of *Clostridiales* abundance and the decrease of *Bacillales* abundance may play an important role in alleviating the allergy induced by soybean protein. It is possible that the microenvironmental pH was lowered due to the production of SCFAs by *Lactobacillus* in the intestine, thereby promoting the growth of *Clostridiales.* At mucosal sites, Tregs may be induced not only by DCs but also possibly by other factors, such as *Clostridia* [45]. Previous studies have shown that different immunoprotective symbiotic bacteria, including *Clostridiales* and *Bacteroidales* members, could induce the expression of transcription factor ROR-γt in neonatal Tregs through a MyD88-dependent mechanism upstream of the Toll-like receptor signaling pathway, thereby enhancing the tolerance of the body-to-food antigens [50]. Atarashi [51] aimed to isolate the strains capable of inducing Tregs from the microbiome to treat human immune diseases. They found that 17 *Clostridial* strains isolated from mice and healthy humans could induce Tregs, and the oral administration of these 17 human-derived strains in adult mice attenuated the disease in a model of allergic diarrhea. It can be speculated that the gavage of *Lactobacillus* in this study may also change the composition of gut microbiota, especially *Clostridia*, thereby inducing Tregs to establish immune tolerance.

Based on the above analysis of intestinal microbiota, it can be found that the richness of the intestinal microbiota (Sobs, Shannon, Ace indexes) of the Lp group is more similar to that of the non-sensitized mice. However, interestingly, the proportion of *Lactobacillales* in the Ld group tended to be higher than that in the SP group (*p* < 0.1), and even higher than that in the normal mice. It is suggested that Ld leads to an increased proportion of *Lactobacillus,* and it may be easier to colonize in the gut. It has been reported that the early colonization of *Lactobacillus delbrueckii* in the intestinal tract of sea bass can stimulate the production of T cells in the mucosa without affecting the integrity of the intestine [52]. When the immune system is stimulated, supplementation with this strain reduces the transcription of proinflammatory genes. Moreover, according to the flow cytometry results of various immune cells, Ld (*Lactobacillus delbrueckii* subsp. *Bulgaricus* CICC 6103) had the best effect in regulating Th1/Th2 to the Th1, and inhibiting Th17 and promoting Tregs differentiation, which made the indexes of the allergic mice more similar to those of normal mice.

## 5. Conclusions

In summary, the positive effects of the three *Lactobacillus* on the intestinal mucosal immune system of soy-allergic mice were illustrated from the aspects of intestinal immune cell differentiation and gut microbiota. It is well known that the increase of Tregs promotes the establishment of immune tolerance. In this study, it was found that the increase of Tregs was positively correlated with the proliferation of DCs and the abundance of *Clostridiales*. Therefore, it is speculated that the establishment of soybean protein immune tolerance after *Lactobacillus* treatment may be achieved through two aspects. In the aspect of immune cells, the Tregs were induced to differentiate by the DCs, which are central to initiating adaptive immunity; in the aspect of gut microbiota, the Tregs were induced to differentiate by *Clostridiales*. In general, the three strains have the ability to reduce the sensitization of soybean, and the order of their ability to alleviate allergy is Ld, La, Lp. According to the results of this study, *Lactobacillus delbrueckii* subsp. *Bulgaricus* CICC 6103 has the potential to develop anti soybean-allergy products. It will be selected as a strain with a better anti-allergic effect as the object of our subsequent research. Furthermore, this study only discussed the modulating effect of exogenous *Lactobacillus* on soy-allergic mice, and further studies are needed to investigate whether ingested *Lactobacillus* can colonize the intestine and interact with gut microbiota.

## Figures and Tables

**Figure 1 foods-12-00627-f001:**
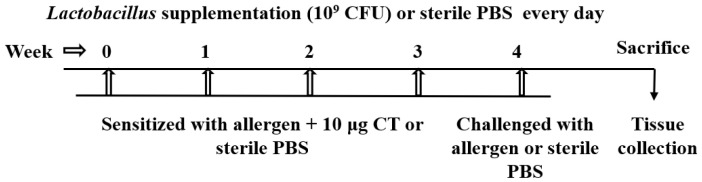
Schedule of sensitization and challenge experiment of BALB/c mice.

**Figure 2 foods-12-00627-f002:**
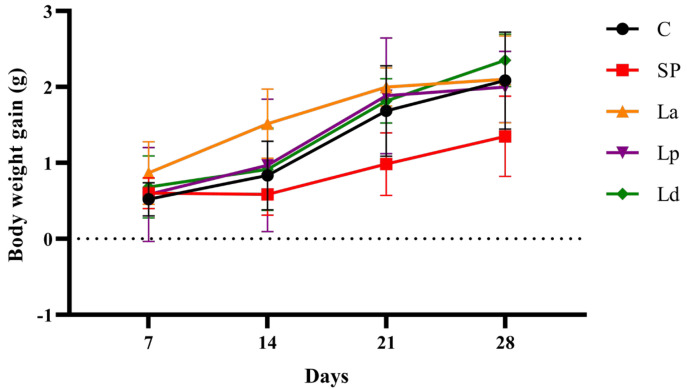
Effects of *Lactobacillus* on body weight gain.

**Figure 3 foods-12-00627-f003:**
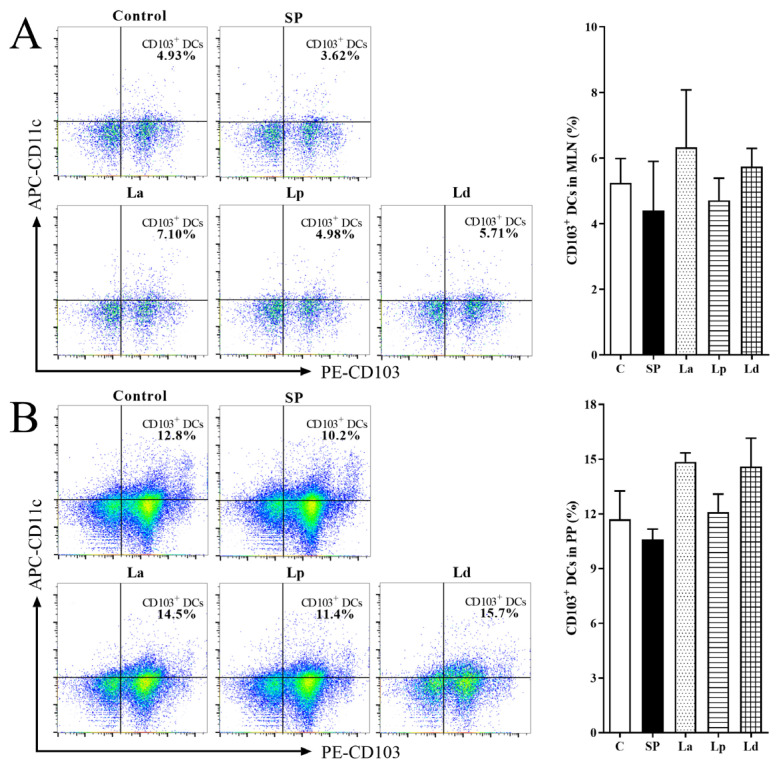
Effect of *Lactobacillus* on the percentage of DCs in the MLNs (**A**) and PPs (**B**) of soy-allergic mice.

**Figure 4 foods-12-00627-f004:**
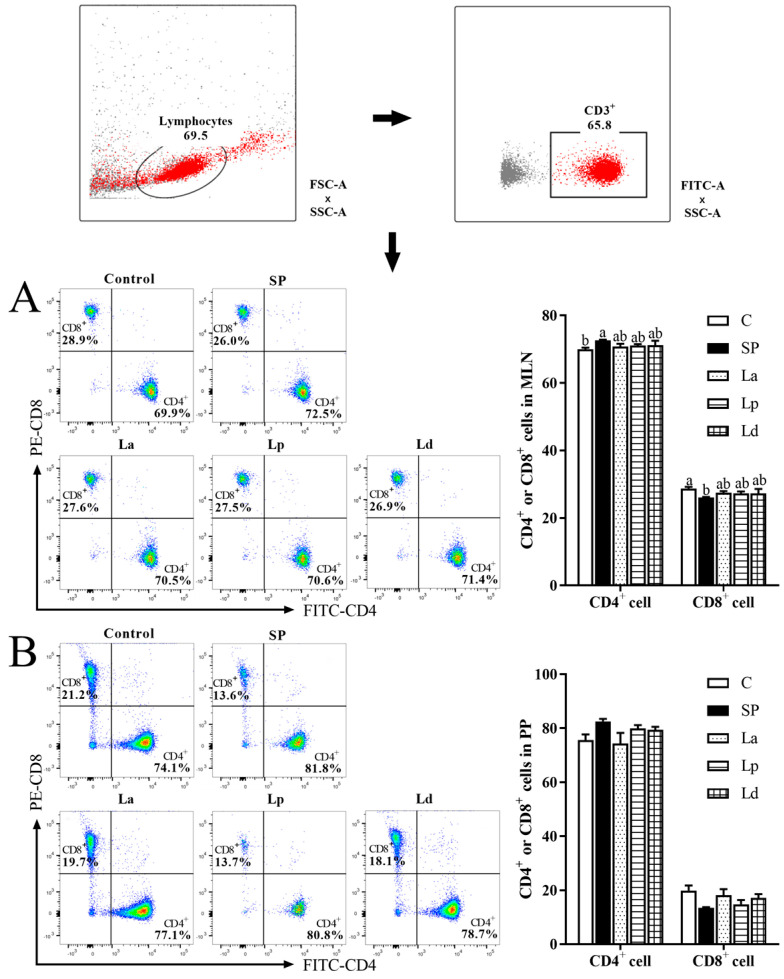
Effect of *Lactobacillus* on the percentage of Th (CD4^+^) and Tc (CD8^+^) cells in the MLNs (**A**) and PPs (**B**) of soy-allergic mice (*p* < 0.05). Different letters denote statistically significant differences (*p* < 0.05).

**Figure 5 foods-12-00627-f005:**
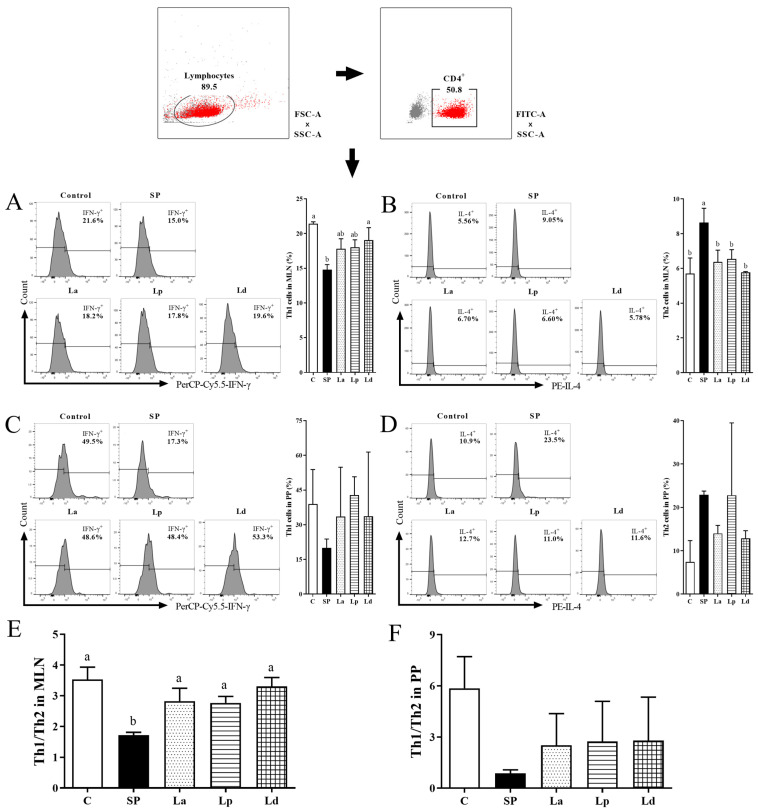
Effect of *Lactobacillus* on Th1/Th2 balance in MLNs and PPs of soy-allergic mice. The graph shows the percentage of Th1 (**A**), Th2 (**B**) cells among CD4^+^ T cells in MLNs, and Th1 (**C**), Th2 (**D**) cells among CD4^+^ T cells in PPs. Th1/Th2 in MLNs (**E**) and PPs (**F**) of mice was analyzed. Different letters denote statistically significant differences (*p* < 0.05).

**Figure 6 foods-12-00627-f006:**
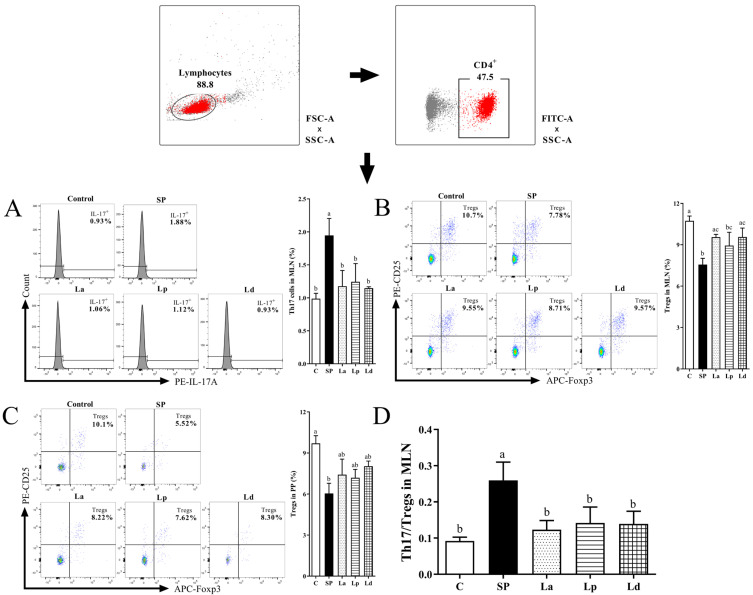
Effect of *Lactobacillus* on Th17/Tregs balance in MLNs and PPs of soy-allergic mice. The graph shows the percentage of Th17 cells among CD4^+^ T cells in MLNs (**A**), the percentage of Tregs among CD4^+^ T in MLNs (**B**) and PPs (**C**) of mice. Th17/Tregs (**D**) in MLNs of mice was analyzed. Different letters denote statistically significant differences (*p* < 0.05).

**Figure 7 foods-12-00627-f007:**
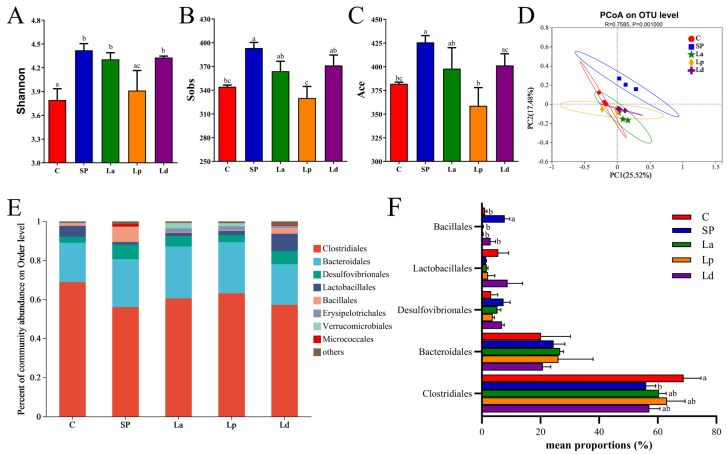
*Lactobacillus* modulates gut microbiota composition in soy-allergic mice. Diversity analysis of the gut microbiota of each group, including Sobs index (**A**), Shannon index (**B**), Ace index (**C**) and PCoA analysis (**D**). Gut microbial community composition (**E**) and inter-group differences in the top five abundances (**F**) at the Order level. Different letters denote statistically significant differences (*p* < 0.05).

**Table 1 foods-12-00627-t001:** Anaphylactic symptom scoring.

Score	Symptoms
0	No symptoms
1	Scratching nose and head
2	Swelling around the eyes and mouth; diarrhea; reduced or stationary activity; accelerating breathing
3	Blue rash around the mouth and tail; labored breathing
4	Loss of consciousness, tremors or seizures
5	Shocking and death

**Table 2 foods-12-00627-t002:** Sensitization scores in different groups of mice.

Groups (n = 6)	Score					
	0	1	2	3	4	5
C	6	0	0	0	0	0
SP	0	1	5	0	0	0
La	4	2	0	0	0	0
Lp	2	4	0	0	0	0
Ld	3	3	0	0	0	0

## Data Availability

All data presented within the article are available upon reasonable request from the corresponding authors.

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
