# Peer review of "Effects of Lactobacillus on the Differentiation of Intestinal Mucosa Immune Cells and the Composition of Gut Microbiota in Soybean-Sensitized Mice"

_foods, 2023, doi:10.3390/foods12030627_

Round 1

Reviewer 1 Report

Regarding MS entitled ‘’ Effects of lactobacillus on the differentiation of intestinal mucosa immune cells and the composition of gut microbiota in soybean sensitized mice’’ I recommended the authors ask help from a native English speaker to double-check their manuscript because there are many grammatical and writing styles errors.

Lactobacillus, the first letter is capitalized and italic, please revise throughout the manuscript.

L14. Define the tree strains.

L20. Please add p-value for significant findings, revise throughout the abstract

L21. The richness and diversity of intestinal microbiota, this sentence is general please be precise in the abstract.

L21-23, please rephrase, difficult to understand

L95. No patients are involved in the study, please be precise and revise the objectives.

L96. Hypothesis is missing

L124. Add ref.

L142. 5 groups (n=6), 6 for what? replicate, please clarify

L145. On which basis, the authors chose this dose?

L155. The effect of Lactobacillus on body weight should be in the results section.

L173. Add ref

L223. The results of body weight should be here.

L237-243; 276-280; and 311-314. These sentences should be in the discussion section. Please move

3.2. and 3.3. please add p-value for significant findings.

Author Response

Response to Reviewer 1 Comments

Point 1: Regarding MS entitled “Effects of lactobacillus on the differentiation of intestinal mucosa immune cells and the composition of gut microbiota in soybean sensitized mice’’ I recommended the authors ask help from a native English speaker to double-check their manuscript because there are many grammatical and writing styles errors.

Response 1: Thank you for your carefully reading. We examined the language carefully, again, and corrected some mistakes.

Point 2: Lactobacillus, the first letter is capitalized and italic, please revise throughout the manuscript.

Response 2: Thank you for your reminding. All “lactobacillus” had been modified to “Lactobacillus” in the revised manuscript.

Point 3: L14. Define the tree strains.

Response 3: Thanks for your suggestion, we had defined the three strains in line 15-17.

Point 4: L20. Please add p-value for significant findings, revise throughout the abstract.

Response 4: Thanks for your reminding, we added p-value for significant findings in line 23-27.

Point 5: L21. The richness and diversity of intestinal microbiota, this sentence is general please be precise in the abstract.

Response 5: Thank for your suggestion. In the revision, we described changes in specific indicators in line 24-25.

Point 6: L21-23, please rephrase, difficult to understand.

Response 6: Thanks for your reminding. We have modified the sentence in line 24-27.

Point 7: L95. No patients are involved in the study, please be precise and revise the objectives.

Response 7: Thanks for your reminding. We had replaced “patients” with “mice” in line 123.

Point 8: L96. Hypothesis is missing.

Response 8: Thanks for your reading. However, there was no hypothesis.

Point 9: L124. Add ref.

Response 9: Thank you for pointing it out, we have added the appropriate reference in lines 155.

Point 10: L142. 5 groups (n=6), 6 for what? replicate, please clarify.

Response 10: Thanks for your reminding. "n=6" means that there were 6 mice per group.

Point 11: L145. On which basis, the authors chose this dose?

Response 11: Thank you for carefully reading. This dose was chosen from references, and we have added the reference in lines 176.

Point 12: L155. The effect of Lactobacillus on body weight should be in the results section.

Response 12: Thanks for your reminding. We had moved the figure to line 273.

Point 13: L173. Add ref.

Response 13: Thanks for your reminding. But there is the section header. Is the line number wrong?

Point 14: L223. The results of body weight should be here.

Response 14: Thanks for your valuable suggestion. We had moved the figure to line 273.

Point 15: L237-243; 276-280; and 311-314. These sentences should be in the discussion section. Please move.

Response 15: Thanks for pointing them out, we had moved the sentence in line 237-238 to line 450-451 of revised manuscript and deleted the sentence in line 238-243.

We had deleted the sentence in line 276-277 because it was duplicated in the introduction section. However, we believe that the sentences 277-280 remain as they are, which can tell the reader why we use the content of IFN-γ / IL-4 to reflect the proportion of Th1 cells / Th2 cells.

We had deleted the sentence in line 311-313 because it was duplicated in the discussion section. However, we believe that the sentences 313-314 remain as they are to inform the reader that IL-17A is a marker of Th17 cells.

Point 16: 3.2. and 3.3. please add p-value for significant findings.

Response 16: Thanks for your reminding, we had added p-value in line 296, 307.

Reviewer 2 Report

Foods

Summary: The manuscript of Yang et al. “Effects of lactobacillus on the differentiation of intestinal mucosa immune cells and the composition of gut microbiota in soybean sensitized mice” investigates the effect of 3 strains of probiotic (Lactobacillus) supplementation in mice affected by soybean food allergy.

General comments:

The topic of the manuscript is interesting and relevant in the field. The manuscript need English editing (overall the verbs and the subject of the verbs) and the experimental design presents some bias. Only few sentences need to be revised. Material & Method section need more details and statistical analysis should be better described.

Specific comments

Abstract:

Line 16: may the authors mean: “…. lactobacillus administered by gavage to soybean allergic mice through the differentiation …..”

Line 20: compare to?

Line 21-23: the sentence is not clear; compared to control group?

Key words:

Add “probiotics”

Introduction:

The authors should deeply introduce the problem of the swine oxidative stress during pregnancy and lactation (the metabolism and the blood parameters). Moreover, the mechanism of action of the amino acid activity as supplementation in diets should be presented. Describe the activity of Ala supplementation alone in pig or other species.

Line 73: replace “that” with “if”

Materials and Methods:

Line 30: replace “allergic” with “allergenic”

Line 33: the sentence is not clear. 2,7% is the average value? And what value is 12,9%?

The authors should briefly describe the symptomatology of soybean allergy in humans, the diagnosis and therapy.

Line 45-51: the sentence is not clear. Which is the molecular mechanism that trigger the allergic reaction? How DCs respond to downregulate the allergic response?

Line 48: replace “integral” with “integrin”

Line 51-53: the sentence is not clear. Please better explain the tolerance mechanism.

Line 54: what does it means “scholar believed”?

Line 58: what does it means “largest immune system”?

Line 64-66: the sentence is not clear. The authors should better explain the sentence.

Line 68: replace “our” with “the”; not only fermented products, but also isolated probiotic strains.

Line 71: replace “regulating” with “regulation”

Line 72: replace “regulate” with “induce”

The authors should deeply describe the hypothesis of activity mechanism of lactobacillus could alleviate food allergies.

L 92: add “genus” after Lactobacillus

Line 96: replace “patients” with “mice”

M & M:

2.1 section: animals are not material; add a paragraph “animals” or include the information on mice in paragraph 2.4.

2.2 section: the authors should mention the exact temperatures they used; what kind of medium was used?

Line 121: replace “solution” with “suspension”

2.3 section: the soybeans were soaked with water (distilled)? Ratio soybean (gr)/water (ml)?

2.4 section: how many days (total) the mice were gavaged with probiotics? Why the authors used second generation of mice? The sensitization was made one per week?

Figure 1B should be moved to Results section.

2.5 From the scheme of the food allergy induction model the animals are sacrificed at 28 days (40 min after the challenge). How the authors weighted the animals at 35 days? Was the scoring made in blind?

Line 165: replace “the solution” with “cell suspension”

Line 167: please define “complete” and “incomplete” medium (FCS, pyruvate, glutamine, …..)

Line 169: how many cells? In which volume? What is it for the stimulation cocktail?

2.6.1 section: it is not clear the scheme of cell stimulation (groups, stimulated, unstimulated,….). The authors should mention the concentration of plated cells. Why some cells were plated in 48 well plates and some in 96 well plates? The authors did not stained the cells with CD45, this is important to monitor the modulation of percentage of the different population in different experimental groups. Why the authors did not prepare a single staining with different colors? How many cells were stained?

2.7 section: how did the authors performed the DNA extraction from caecal content?

2.8 section: did the authors evaluate the distribution of the analysed parameters? Normal or not normal? In case, with what kind of test? Based on the type of distribution of the parameters, different statistical test should be applied (parametric such as ANOVA or non parametric….).

Results:

Line 225: the sentence should be moved to discussion

Line 229: replace “to varying” with “with different”

Line 232: the sentence should be moved to discussion

Line 243: add “of” after “endocytosis”

Line 245: the authors means % or total number of cells? It is not clear if the differences among groups are statistically significant or not. When the differences are significant the authors should mention in the text the average values and the p value. In M&M section the authors sad that they used ANOVA test, but here they compare only 2 groups.

Line 248-50: the sentence is not clear.

Line 251-52: the sentence is not clear. La and Ld have a better effect compared to Lp? Are the differences statistically significant or not?

Line 263-65: the sentence should be moved to discussion

Line 267-70: the sentence should be moved to discussion

The figure caption should be more descriptive and complete (n of animal per group, statistic test, type of experimental analysis, meaning of the letters for the significativity). For example what is the plot in the upper part?

Line 285: compared to? Please mention the %.

Line 300: what does it mean “elevated”? It means a tendency? If so, please define the tendency (for example p<0.1)

Line 300-02: the sentence should be moved to discussion

Line 311-12: the sentence should be moved to discussion

Line 315-17: the sentence should be moved to discussion

Line 322 why “obviously”?

Line 324: replace “When the ….” With “In allergic subjects,….”

Line 329: replace “significant” with “significantly”

All over the manuscript the authors use “proportion” but should be more correct “percentage” or “frequency”.

Line 331-33: the sentence should be moved to discussion

Line 341-43: the sentence should be moved to discussion

Line 356: replace “five groups mice…” with “the five experimental groups….”

Line 367: replace “intestine” with “caecum”

Discussion:

The authors should add at the beginning of this section an introduction sentence.

Line 396: what does it means “primer immunity”? The sentence is not clear.

Line 400-01: the authors should introduce the retinoic acid signaling and its involvement with probiotic activity.

Line 405: add “was” before “found”

Line 409: replace “have shown” whit “showed”

Line 429: there is a repetition of “production”

Line 470-71: the sentence is not clear

Line 472-73: the sentence is not clear

Line 474: may be the authors means “sea bass” instead of “weever”?

Line 476: what “strain”? the sentence is not clear

Line 480: the sentence should be moved to Conclusions section

Author Response

Response to Reviewer 2 Comments

Point 1: General comments: The topic of the manuscript is interesting and relevant in the field. The manuscript need English editing (overall the verbs and the subject of the verbs) and the experimental design presents some bias. Only few sentences need to be revised. Material & Method section need more details and statistical analysis should be better described.

Response 1: Thank you for your carefully reading. We examined the language carefully, again, and corrected some mistakes.

Point 2: Abstract: Line 16: may the authors mean: “…. lactobacillus administered by gavage to soybean allergic mice through the differentiation …..”

Response 2: Thank you for your reminding. We had modified the sentence in line 18.

Point 3: Line 20: compare to?

Response 3: Thanks for your reminding. We had modified the sentence and added p-value in line 22-23.

Point 4: Line 21-23: the sentence is not clear; compared to control group?

Response 4: Thank you for pointing it out. We had modified the sentence and added “compared to control group” in line 23-27.

Point 5: Key words: Add “probiotics”

Response 5: Thanks for your suggestion. In the revision, we added the key word "probiotics", seen in Line 30.

Point 6: The authors should deeply introduce the problem of the swine oxidative stress during pregnancy and lactation (the metabolism and the blood parameters). Moreover, the mechanism of action of the amino acid activity as supplementation in diets should be presented. Describe the activity of Ala supplementation alone in pig or other species.

Line 73: replace “that” with “if”.

Response 6: Thanks for your reading. However, there was no “pig” in our manuscript, and we didn't have "that" in line 73. Is it about another manuscript?

Point 7: Line 30: replace “allergic” with “allergenic”.

Response 7: Thanks for your reminding. We had replaced “allergic” with “allergenic” in line 34.

Point 8: Line 33: the sentence is not clear. 2.7% is the average value? And what value is 12.9%?

The authors should briefly describe the symptomatology of soybean allergy in humans, the diagnosis and therapy.

Response 8: Thanks for your valuable suggestions. The average value is 2.7% and the maximum value is 12.9%. In the revision, we modified the sentence in line 37-40. And we added briefly describe the symptomatology of soybean allergy in humans, the diagnosis and therapy in line 35-37, 40-45.

Point 9: Line 45-51: the sentence is not clear. Which is the molecular mechanism that trigger the allergic reaction? How DCs respond to downregulate the allergic response?

Response 9: Thanks for your pointing them out. In the revision, we added some sentences to answer your question, seen in line 55-66.

Point 10: Line 48: replace “integral” with “integrin”.

Response 10: Thanks for your reminding. We had replaced“integral” with “integrin” in line 66.

Point 11: Line 51-53: the sentence is not clear. Please better explain the tolerance mechanism.

Response 11: Thank you for your advice. In the revision, we had modified some sentences to explain the tolerance mechanism, seen in line 66-71.

Point 12: Line 54: what does it means “scholar believed”?

Response 12: Thanks for your reminding. The original meaning was “Some scholars thought”. We had modified the sentence in line 76.

Point 13: Line 58: what does it means “largest immune system”?

Response 13: Thanks for your carefully reading. The intestinal mucosal immune system includes diffuse lymphoid tissue, solitary lymphoid nodule, aggregated lymphoid nodules, lymphocytes, macrophages and plasma cells, etc., the distribution is wide, so it is called "largest immune system".

Point 14: Line 64-66: the sentence is not clear. The authors should better explain the sentence.

Response 14: Thanks for your pointing it out. We had modified the sentence in line 86-87.

Point 15: Line 68: replace “our” with “the”; not only fermented products, but also isolated probiotic strains.

Response 15: Thanks for your valuable suggestion. We had replaced “our” with “the” and added “isolated probiotic strains” has been added in line 91-92.

Point 16: Line 71: replace “regulating” with “regulation”.

Response 16: Thanks for your carefully reading. In the sentence, "regulate the host immune system" after "In terms of" is a verb phrase, so "regulating" should be used.

Point 17: Line 72: replace “regulate” with “induce”

Response 17: Thanks for your reminding. We had replaced “regulate” with “induce” in line 96.

Point 18: The authors should deeply describe the hypothesis of activity mechanism of lactobacillus could alleviate food allergies.

Response 18: Thanks for your reminding. Since the exact mechanism of that Lactobacillus alleviate food allergies is still a mystery, we had listed only a few possibilities in line 97-100.

Point 19: L 92: add “genus” after Lactobacillus

Response 19: Thanks for your reminding. We had added “genus” after “Lactobacillus” in line 120.

Point 20: Line 96: replace “patients” with “mice”

Response 20: Thanks for your pointing it out. We had replaced “patients” with “mice” in line 123.

Point 21: 2.1 section: animals are not material; add a paragraph “animals” or include the information on mice in paragraph 2.4.

Response 21: Thanks for your reminding. In the revision, we include the information on mice in line 162-163.

Point 22: 2.2 section: the authors should mention the exact temperatures they used; what kind of medium was used?

Response 22: Thanks for your carefully reading. In the revision, we mentioned the exact temperatures and the kind of medium was used, seen in line 144-145.

Point 23: Line 121: replace “solution” with “suspension”

Response 23: Thanks for your pointing it out. We had replaced “solution” with “suspension” in line 150.

Point 24: 2.3 section: the soybeans were soaked with water (distilled)? Ratio soybean (gr)/water (ml)?

Response 24: Thanks for your reminding. We had added this detail in line 153.

Point 25: 2.4 section: how many days (total) the mice were gavaged with probiotics? Why the authors used second generation of mice? The sensitization was made one per week?

Response 25: Thanks for your carefully reading. We gavaged with Lactobacillus daily for a total of 28 days. The first generation of mice had been generally fed a common diet containing soy protein. To avoid the production of specific antibodies in the mice, the second generation of mice were fed a special diet without soy meal. The sensitization was made one per week. You can see the details in line 171-181.

Point 26: Figure 1B should be moved to Results section.

Response 26: Thanks for your reminding. We had moved the figure to line 272.

Point 27: 2.5 From the scheme of the food allergy induction model the animals are sacrificed at 28 days (40 min after the challenge). How the authors weighted the animals at 35 days? Was the scoring made in blind?

Response 27: Thanks for your carefully reading. We made a mistake and had corrected it in line 190. And we scored according to Table 1 in line 193.

Point 28: Line 165: replace “the solution” with “cell suspension”

Response 28: Thanks for your pointing out. We had replaced “the solution” with “cell suspension” in line 197.

Point 29: Line 167: please define “complete” and “incomplete” medium (FCS, pyruvate, glutamine, …..)

Response 29: Thanks for your reminding. We had added this detail in line 200-201.

Point 30: Line 169: how many cells? In which volume? What is it for the stimulation cocktail?

Response 30: Thanks for your reminding. We had added count or volume in line 203, 209, 219, 228.The Cell Stimulation Cocktail (plus protein transport inhibitors) (500X) is a cocktail of phorbol 12-myristate 13-acetate (PMA), ionomycin, brefeldin A and monensin. It can be used for induction and subsequent intracellular detection of cytokines and other secreted proteins in both in vitro-cultured and ex vivo cells. Because there is no surface marker to distinguish Th1, Th2 and Th17, we used the content of intracellular characteristic cytokines IFN-γ to reflect the expression of Th1 cells, IL-4 to reflect the content of Th2 cells, IL17A to reflect the expression of Th17 cells. In this paper, we used stimulated cells to detect cytokines, thereby indirectly reflecting cell content.

Point 31: 2.6.1 section: it is not clear the scheme of cell stimulation (groups, stimulated, unstimulated,….). The authors should mention the concentration of plated cells. Why some cells were plated in 48 well plates and some in 96 well plates? The authors did not stained the cells with CD45, this is important to monitor the modulation of percentage of the different population in different experimental groups. Why the authors did not prepare a single staining with different colors? How many cells were stained?

Response 31: Thank you for carefully reading, only a part of cells was stimulated in each group. These cells needed to be stimulated for 16h before detecting Th1, Th2, and Th17 cells, so they were plated in 48-well cell culture plates in an incubator and then stained in 96-well tip plates after stimulation. However, other cells were used to detect DCs and CD4+/CD8+ T cells, which did not require stimulating cells and could be directly stained, so they were spread on 96-well tip plates.

Thank you for reminding us that we will stain CD45 in future experiments.

We had prepared single staining with different colors. We utilized Anti-Rat and Anti-Hamster Igκ/Negative Control Compensation Particles Set to adjust compensation.

You can see the number of stained cells in line 209, 219, 228.

Point 32: 2.7 section: how did the authors performed the DNA extraction from caecal content?

Response 32: Thanks for your reminding. Since this part of the experiment was commissioned by the company, we did not know which method to use.

Point 33: 2.8 section: did the authors evaluate the distribution of the analysed parameters? Normal or not normal? In case, with what kind of test? Based on the type of distribution of the parameters, different statistical test should be applied (parametric such as ANOVA or non parametric….).

Response 33: Thanks for your valuable suggestion. We had applied parameters tests. Firstly, GraphPad Prism 8 was used to perform normality tests and check by Shapiro-Wilk test. Then, based on testing, one-way ANOVA was used, and Tukey test was used for multiple comparisons between the different treatment groups. We had modified the section in line 252-259.

Point 34: Results: Line 225: the sentence should be moved to discussion

Response 34: Thanks for your reminding. We had deleted the sentence, because it was duplicated in the discussion section, seen in line 448.

Point 35: Line 229: replace “to varying” with “with different”

Response 35: Thanks for your pointing it out. We had replaced“to varying” with “with different” in line 267.

Point 36: Line 232: the sentence should be moved to discussion

Response 36: Thanks for your reminding. We had moved the sentence to discussion in line 446-449.

Point 37: Line 243: add “of” after “endocytosis”

Response 37: Thanks for your reminding. We had deleted the sentence.

Point 38: Line 245: the authors means % or total number of cells? It is not clear if the differences among groups are statistically significant or not. When the differences are significant the authors should mention in the text the average values and the p value. In M&M section the authors sad that they used ANOVA test, but here they compare only 2 groups.

Response 38: Thanks for your reminding. This "%" means the percentage of positive cells in lymphocytes.

There were no significant differences between the groups in figure 3, seen in line 286.

We had added the p-value when the differences are significant in manuscript in line 307, 350 etc..

We used ANOVA test. And we compared other groups in line 290-296.

Point 39: Line 248-50: the sentence is not clear.

Response 39: Thanks for your pointing it out. We had modified the sentence in line 291-292.

Point 40: Line 251-52: the sentence is not clear. La and Ld have a better effect compared to Lp? -Are the differences statistically significant or not?

Response 40: Thanks for your reminding. We had added the p-value in line 296, but there were no significant differences between the groups.

Point 41: Line 263-65: the sentence should be moved to discussion

Response 41: Thanks for your reminding. But we thought that is a part of the result.

Point 42: Line 267-70: the sentence should be moved to discussion

Response 42: Thanks for your pointing it out. We deleted the sentence in line 315. But we thought that the sentence in line 312-314 is a part of the result.

Point 43: The figure caption should be more descriptive and complete (n of animal per group, statistic test, type of experimental analysis, meaning of the letters for the significativity). For example what is the plot in the upper part?

Response 43: Thanks for your pointing it out. “n of animal per group” can been seen in line 173, and “statistic test, type of experimental analysis, meaning of the letters for the significativity” can been seen in line 252-259.

Point 44: Line 285: compared to? Please mention the %.

Response 44: Thanks for your suggestions. We added “compared to SP group” and mentioned the percentage of each group in line 332-333.

Point 45: Line 300: what does it mean “elevated”? It means a tendency? If so, please define the tendency (for example p<0.1)

Response 45: Thanks for your pointing it out. We had changed “elevated” to “slightly increased” in line 350, it didn’t mean a tendency or have significant different.

Point 46: Line 300-02: the sentence should be moved to discussion

Response 46: Thanks for your reminding. We had moved the sentence to line 487-491.

Point 47: Line 311-12: the sentence should be moved to discussion

Response 47: Thanks for your reminding. We had deleted the sentence because it was duplicated in the discussion section.

Point 48: Line 315-17: the sentence should be moved to discussion

Response 48: Thanks for your reminding. But we thought that is a part of the result.

Point 49: Line 322 why “obviously”?

Response 49: Thanks for your carefully reading. Ld decreased the proportion of CD4+ IL-17A+ cells in MLN of soybean protein-sensitized mice the most among the three strains (from 1.95% to 1.15%), as shown in line 371,373, so this used“obviously”.

Point 50: Line 324: replace “When the ….” With “In allergic subjects,….”

Response 50: Thanks for your pointing it out. We had replaced “When the body is allergic” with “In allergic subjects,” in line 376.

Point 51: Line 329: replace “significant” with “significantly”

Response 51: Thanks for your suggestion. We had replaced “significant” with “significantly” in line 379.

Point 52: All over the manuscript the authors use “proportion” but should be more correct “percentage” or “frequency”.

Response 52: Thanks for your pointing it out. In terms of cell number, we had replaced “proportion” with “percentage”, but in terms of gut microbiota, we still thought "proportion" is more appropriate.

Point 53: Line 331-33: the sentence should be moved to discussion

Response 53: Thanks for your reminding. We had moved the sentence to line 491-492.

Point 54: Line 341-43: the sentence should be moved to discussion

Response 54: Thanks for your reminding. We had moved the sentence to line 493-494.

Point 55: Line 356: replace “five groups mice…” with “the five experimental groups….”

Response 55: Thanks for your pointing it out. We had replaced “five groups mice” with “the five experimental groups” in line 410-411.

Point 56: Line 367: replace “intestine” with “caecum”

Response 56: Thanks for your pointing it out. We had replaced “intestine” with “caecum” in line 422.

Point 57: Discussion: The authors should add at the beginning of this section an introduction sentence.

Response 57: Thanks for your advice. In the revision, we added some sentences at the beginning of this section in line 445-451.

Point 58: Line 396: what does it means “primer immunity”? The sentence is not clear.

Response 58: Thanks for your reminding. We had modified the sentence in line 459.

Point 59: Line 400-01: the authors should introduce the retinoic acid signaling and its involvement with probiotic activity.

Response 59: Thanks for your reminding. In the revision, we introduced it in line 62-66.

Point 60: Line 405: add “was” before “found”

Response 60: Thanks for your pointing it out. In the revision, we added “was” before “found”, seen in line 468.

Point 61: Line 409: replace “have shown” with “showed”

Response 61: Thanks for your suggestion. We had replaced “have shown” with “showed” in line 472.

Point 62: Line 429: there is a repetition of “production”

Response 62: Thanks for your carefully reading. In the revision, we deleted the second “production” in line 500.

Point 63: Line 470-71: the sentence is not clear

Response 63: Thanks for your advice. We had modified the sentence in line 543-544.

Point 64: Line 472-73: the sentence is not clear

Response 64: Thanks for your reminding. We had modified the sentence in line 546-548.

Point 65: Line 474: may be the authors means “sea bass” instead of “weever”?

Response 65: Thanks for your carefully reading. We had replaced “weever” with “sea bass” in line 550.

Point 66: Line 476: what “strain”? the sentence is not clear

Response 66: Thanks for your reminding. We had modified the sentence in line 552.

Point 67: Line 480: the sentence should be moved to Conclusions section

Response 67: Thanks for your pointing it out. We had moved the sentence in line 572-573.

Reviewer 3 Report

It is well known that soybean is one of the allergens of food allergy, and this reviewer appreciates the clinical value of this study, which examined the basic mechanism of probiotics against food allergy. However, I think some corrections should be needed.

Please describe the resistance of the three types of Lactobacillus used in this study to stomach acid.

The authors should describe why lactobacillus administration increased the number of bacteria of Order Clostridiales. Clostridium administration is more useful?
Also, wouldn't clostridium administration be more useful than lactobacillus administration?

It is reported that the diversity of gut microbiota is decreased in atopic dermatitis and allergic rhinitis, while it is increased in food allergy in this study. The authors should explain this discrepancy to help the reader understand.

Round 2

Reviewer 1 Report

Dear authors,

Thank you very much for your revisions.